# Turbo Autoencoder: Deep learning based channel codes for point-to-point communication channels

**Yihan Jiang**
ECE Department
University of Washington
Seattle, United States
yij021@uw.edu

**Hyeji Kim**
Samsung AI Center
Cambridge
Cambridge, United
Kingdom
hkim1505@gmail.com

**Himanshu Asnani**
School of Technology
and Computer Science
Tata Institute of
Fundamental Research
Mumbai, India
himanshu.asnani@tifr.res.in

**Sreeram Kannan**
ECE Department
University of Washington
Seattle, United States
ksreeram@ee.washington.edu

**Sewoong Oh**
Allen School of
Computer Science &
Engineering
University of Washington
Seattle, United States
sewoong@cs.washington.edu

**Pramod Viswanath**
ECE Department
University of Illinois at
Urbana Champaign
Illinois, United States
pramodv@illinois.edu

## Abstract

Designing codes that combat the noise in a communication medium has remained a significant area of research in information theory as well as wireless communications. Asymptotically optimal channel codes have been developed by mathematicians for communicating under canonical models after over 60 years of research. On the other hand, in many non-canonical channel settings, optimal codes do not exist and the codes designed for canonical models are adapted via heuristics to these channels and are thus not guaranteed to be optimal. In this work, we make significant progress on this problem by designing a fully end-to-end jointly trained neural encoder and decoder, namely, Turbo Autoencoder (TurboAE), with the following contributions: (a) under moderate block lengths, TurboAE approaches state-of-the-art performance under canonical channels; (b) moreover, TurboAE outperforms the state-of-the-art codes under non-canonical settings in terms of reliability. TurboAE shows that the development of channel coding design can be automated via deep learning, with near-optimal performance.

## 1 Introduction

Autoencoder is a powerful unsupervised learning framework to learn latent representations by minimizing reconstruction loss of the input data [1]. Autoencoders have been widely used in unsupervised learning tasks such as representation learning [1] [2], denoising [3], and generative model [4] [5]. Most autoencoders are under-complete autoencoders, for which the latent space is smaller than the input data [2]. Over-complete autoencoders have latent space larger than input data. While the goal of under-complete autoencoder is to find a low dimensional representation of input data, the goal of over-complete autoencoder is to find a higher dimensional representation of input data so that from a noisy version of the higher dimensional representation, original data can be reliably recovered. Over-complete autoencoders are used in sparse representation learning [3] [6] and robust representation learning [7].

Channel coding aims at communicating a message over a noisy random channel [8]. As shown in Figure 1 left, the transmitter maps a message to a codeword via adding redundancy (this mapping is called encoding). A channel between the transmitter and the receiver randomly corrupts the codeword so that the receiver observes a noisy version which is used by the receiver to estimate the transmitted

message (this process is called decoding). The encoder and the decoder together can be naturally viewed as an over-complete autoencoder, where the noisy channel in the middle corrupts the hidden representation (codeword). Therefore, designing a reliable autoencoder can have a strong bearing on alternative ways of designing new encoding and decoding schemes for wireless communication systems.

Traditionally, the design of communication algorithms first involves designing a 'code' (i.e., the encoder) via optimizing certain mathematical properties of encoder such as minimum code distance [9]. The associated decoder that minimizes the bit-error-rate then isderived based on the maximum a posteriori (MAP) principle. However, while the optimal MAP decoder is computationally simple for some simple codes (e.g., convolutional codes), for known capacity-achieving codes, the MAP decoder is not computationally efficient; hence, alternative decoding principles such as belief propagation are employed (e.g., for decoding turbo codes). The progress on the design of optimal channel codes with computationally efficient decoders has been quite sporadic due to its reliance on human ingenuity. Since Shannon's seminal work in 1948 [8], it took several decades of research to finally reach to the current state-of-the-art codes [10].

Near-optimal channel codes such as Turbo [11], Low Density Parity Check (LDPC) [12], and Polar codes [10] show Shannon capacity-approaching [8] performance on AWGN channels, and they have had a tremendous impact on the Long Term Evolution (LTE) and 5G standards. The traditional approach has the following caveats:

($a$) Decoder design heavily relies on handcrafted optimal decoding algorithms for the canonical Additive White Gaussian Noise (AWGN) channels, where the signal is corrupted by i.i.d. Gaussian noise. In practical channels, when the channel deviates from AWGN settings, often times heuristics are used to compensate the non-Gaussian properties of the noise, which leaves a room for the potential improvement in reliability of a decoder [9] [13].

($b$) Channel codes are designed for a finite block length $K$. Channel codes are guaranteed to be optimal only when the block-length approaches infinity, and thus are near-optimal in practice only when the block-length is large. On the other hand, under short and moderate block length regimes, there is a room for improvement [14].

($c$) The encoder designed for the AWGN channel is used across a large family of channels, while the decoder is adapted. This design methodology fails to utilize the flexibility of the encoder.

**Related work.** Deep learning has pushed the state-of-the-art performance of computer vision and natural language processing to a new level far beyond handcrafted algorithms in a data-driven fashion [15]. There also has been a recent movement in applying deep learning to wireless communications. Deep learning based channel decoder design has been studied since [16] [17], where encoder is fixed as a near-optimal code. It is shown that belief propagation decoders for LDPC and Polar codes can be imitated by neural networks [18] [19] [20] [21] [22]. It is also shown that convolutional and turbo codes can be decoded optimally via Recurrent Neural Networks (RNN) [23] and Convolutional Neural Networks (CNN) [24]. Equipping a decoder with a learnable neural network also allows fast adaptation via meta-learning [25]. Recent works also extend deep learning to multiple-input and multiple-output (MIMO) settings [26]. While *neural decoders* show improved performance on various communication channels, there has been limited success in inventing novel codes using this paradigm. Training methods for improving both modulation and channel coding are introduced in [16] [17], where a (7,4) neural code mapping a 4-bit message to a length-7 codeword can match (7,4) Hamming code performance. Current research includes training an encoder and a decoder with noisy feedback [27], improving modulation gain [28], as well as extensions to multi-terminal settings [29]. Joint source-channel coding shows improved results combining source coding (compression) along with channel coding (noise mitigation) [30]. Neural codes were shown to outperform existing state-of-the-art codes on the feedback channel [31]. However, in the canonical setting of AWGN channel, neural codes are still far from capacity-approaching performance due to the following challenges.

(Challenge A) Encoding with randomness is critical to harvest coding gain on long block lengths [8]. However, existing sequential neural models, both CNN and even RNN, can only learn limited local dependency [32]. Hence, neural encoder cannot sufficiently utilize the benefits of even moderate block length.

(Challenge B) Training neural encoder and decoder jointly (with a random channel in between) introduces optimization issues where the algorithm gets stuck at local optima. Hence, a novel training algorithm is needed.

**Contributions.** In this paper, we confront the above challenges by introducing Turbo Autoencoder (henceforth, TurboAE) – the first channel coding scheme with both encoder and decoder powered by neural networks that achieves reliability close to the state-of-the-art channel codes under AWGN channels for a moderate block length. We demonstrate that channel coding, which has been a focus of study by mathematicians for several decades [9], can be learned in an end-to-end fashion from data alone. Our major contributions are:

- We introduce TurboAE, a neural network based over-complete autoencoder parameterized as Convolutional Neural Networks (CNN) along with interleavers (permutation) and de-interleavers (de-permutation) inspired by turbo codes (Section 3.1). We introduce TurboAE-binary, which binarizes the codewords via straight-through estimator (Section 3.2).

- We propose techniques that are critical for training TurboAE which includes mechanisms of alternate training of encoder and decoder as well as strategies to choose right training examples. Our training methodology ensures stable training of TurboAE without getting trapped at locally optimal encoder-decoder solutions. (Section 3.3)

- Compared to multiple capacity-approaching codes on AWGN channels, TurboAE shows superior performance in the low to middle SNR range when the block length is of moderate size ($K \sim 100$). To the best of our knowledge, this is the first result demonstrating the deep learning powered discovered neural codes can outperform traditional codes in the canonical AWGN setting (Section 4.1).

- On a non-AWGN channel, fine-tuned TurboAE shows significant improvements over state-of-the-art coding schemes due to the flexibility of encoder design, which shows that TurboAE has advantages on designing codes where handcrafted solutions fail (Section 4.2).

## 2   Problem Formation

The channel coding problem is illustrated in Figure 1 left, which consists of three blocks – an encoder $f_\theta(\cdot)$, a channel $c(\cdot)$, and a decoder $g_\phi(.)$. A channel $c(\cdot)$ randomly corrupts an input $x$ and is represented as a probability transition function $p_{y|x}$. A canonical example of channel $c(\cdot)$ is an identically and independently distributed (i.i.d.) AWGN channel, which generates $y_i = x_i + z_i$ for $z_i \sim N(0, \sigma^2)$, $i = 1, \cdots, K$. The encoder $x = f_\theta(u)$ maps a random binary message sequence $u = (u_1, \cdots, u_K) \in \{0,1\}^K$ of block length $K$ to a codeword $x = (x_1, \cdots, x_N)$ of length N, where $x$ must satisfy either soft power constraint where $E(x) = 0$ and $E(x^2) = 1$, or hard power constraint $x \in \{-1, +1\}$. Code rate is defined as $R = \frac{K}{N}$, where $N > K$. The decoder $g_\phi(y)$ maps a real valued received sequence $y = (y_1, \cdots, y_N) \in \mathcal{R}^N$ to an estimate of the transmitted message sequence $\hat{u} = (\hat{u}_1, \cdots, \hat{u}_K) \in \{0,1\}^K$.

AWGN channel allows closed-form mathematical analysis, which has remained as the major playground for channel coding researchers. The noise level is defined as signal-to-noise ratio, $SNR = -10 \log_{10} \sigma^2$. The decoder recovers the original message as $\hat{u} = g_\phi(y)$ using the received signal $y$.

Channel coding aims to minimize the error rate of recovered message $\hat{u}$. The standard metrics are bit error rate (BER), defined as $BER = \frac{1}{K} \sum_1^K \Pr(\hat{u}_i \neq u_i)$, and block error rate (BLER), defined as $BLER = \Pr(\hat{u} \neq u)$.

While canonical capacity-approaching channel codes work well as block length goes to infinity, when the block length is short, they are not guaranteed to be optimal. We show the benchmarks on block length 100 in Figure 1 right with widely-used LDPC, Turbo, Polar, and Tail-bitting Convolutional Code (TBCC), generated via Vienna 5G simulator [33] [34], with code rate $1/3$.

Naively applying deep learning models by replacing encoder and decoder with general purpose neural network does not perform well. Direct applications of fully connected neural network (FCNN) cannot scale to a longer block length; the performance of FCNN-AE is even worse than repetition code [35]. Direct applications where both the encoder and the decoder are Convolutional Autoencoder (termed as CNN-AE [36]) shows better performance than TBCC, but are far from capacity-approaching codes

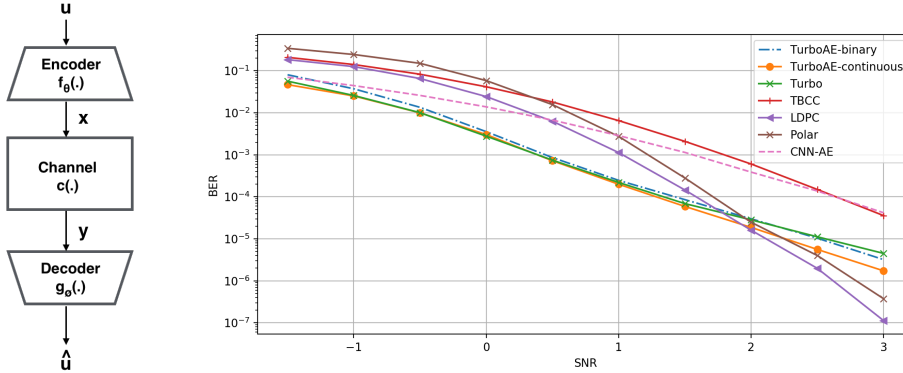

Figure 1: Channel coding can be viewed as an over-complete autoencoder with channel in the middle (left). TurboAE performs well under moderate block length in low and middle SNR (right).

such as LDPC, Polar, and Turbo. Bidirectional RNN and LSTM [35] has similar performance as CNN-AE and is not shown in the figure for clarity. Thus neither CNN nor RNN based auto-encoders can directly approach state-of-the-art performance. A key reason for their shortcoming is that they have only local memory, the encoder only remembers information locally. To have high protection against channel noise, it is necessary to have long term memory.

We propose TurboAE with interleaved encoding and iterative decoding that creates long term memory in the code and shows a significant improvement compared to CNN-AE. TurboAE has two versions, TurboAE-continuous which faces soft power constraint (i.e., the total power across a codeword is bounded) and TurboAE-binary which faces hard power constraint (i.e., each transmitted symbol has a power constraint - and is thus forced to be binary). Both TurboAE-binary and TurboAE-continuous perform comparable or better than all other capacity-approaching codes at a low SNR, while at a high SNR (over 2 dB with $BER < 10^{-5}$), the performance is only worse than LDPC and Polar code.

## 3   TurboAE : Architecture Design and Training

### 3.1   Design of TurboAE

**Turbo code and turbo principle**: Turbo code is the first capacity-approaching code ever designed [11]. There are two novel components of Turbo code which led to its success: an interleaved encoder and an iterative decoder. The starting point of the Turbo code is a recursive systematic convolutional (RSC) code which has an optimal decoding algorithm (the Bahl-Cocke-Jelinek-Raviv (BCJR) algorithm [37]). A key disadvantage in the RSC code is that the algorithm lacks long range memory (since the convolutional code operates on a sliding window). The key insight of Berrou was to introduce long range memory by creating two copies of the input bits - the first goes through the RSC code and the second copy goes through an interleaver (which is a permutation of the bits) before going through the same code. Such a code can be decoded by iteratively alternating between soft-decoding based on the signal received from the first copy and then using the de-interleaved version as a prior to decode the second copy. The 'Turbo principle' [38] refers to the iterative decoding with successively refining the posterior distribution on the transmitted bits across decoding stages with original and interleaved order. This code is known to have excellent performance, and inspired by this, we design TurboAE featuring both learnable interleaved encoder and iterative decoder.

**Interleaved Encoding Structure**: Interleaving is widely used in communication systems to mitigate bursty noise [39]. Formally, interleaver $x^\pi = \pi(x)$ and de-interleaver $x = \pi^{-1}(x^\pi)$ shuffle and shuffle back the input sequence $x$ with the a pseudo random interleaving array known to both encoder and decoder, respectively, as shown in Figure 2 left. In the context of Turbo code and TurboAE, the interleaving is not used to mitigate bursty errors (since we are mainly concerned with i.i.d. channels) but rather to add long range memory in the structure of the code.

We take code rate 1/3 as an example for interleaved encoder $f_\theta$, which consists of three learnable encoding blocks $f_{i,\theta}(.)$ for $i \in \{1, 2, 3\}$, where $f_{i,\theta}(.)$ encodes $b_i = f_\theta(u), i \in \{1, 2\}$ and $b_3 =$

$f_{3,\theta}(\pi(u))$, where $b_i$ is a continuous value. The power constraint of channel coding is enforced via power constraint block $x_i = h(b_i)$.

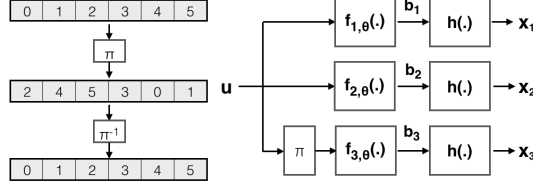

Figure 2: Visualization of Interleaver ($\pi$) and De-interleaver ($\pi^{-1}$) (left); TurboAE encoder on code rate 1/3 (right)

**Iterative Decoding Structure**: As received codewords are encoded from original message $u$ and interleaved message $\pi(u)$, decoding interleaved code requires iterative decoding on both interleaved and de-interleaved order shown in Figure 3. Let $y_1, y_2, y_3$ denote noisy versions of $x_1, x_2, x_3$, respectively. The decoder runs multiple iterations, with each iteration contains two decoders $g_{\phi_{i,1}}$ and $g_{\phi_{i,2}}$ for interleaved and de-interleaved order on the $i$-th iteration.

The first decoder $g_{\phi_{i,1}}$ takes received signal $y_1$, $y_2$ and de-interleaved prior $p$ with shape $(K, F)$, where as $F$ is the information feature size for each code bit, to produce the posterior $q$ with same shape $(K, F)$. The second decoder $g_{\phi_{i,2}}$ takes interleaved signal $\pi(y_1)$, $y_3$ and interleaved prior $p$ to produce posterior $q$. The posterior of previous stage $q$ serves as the prior of next stage $p$. The first iteration takes 0 as a prior, and at last iteration the posterior is of shape $(K, 1)$, are decoded as by sigmoid function as $\hat{u} = sigmoid(q)$.

Both encoder and decoder structure can be considered as a parametrization of Turbo code. Once we parametrize the encoder and the decoder, since the encoder, channel, and decoder are differentiable, TurboAE can be trained end-to-end via gradient descent and its variants.

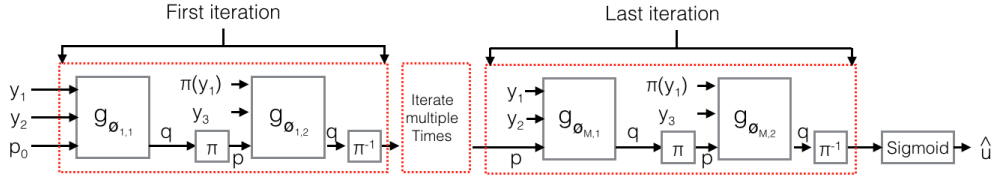

Figure 3: TurboAE iterative decoder on code rate 1/3

**Encoder and Decoder Design**: The space of messages and codewords are exponential (For a length-$K$ binary sequence, there are $2^K$ distinct messages). Hence, the encoder and decoder must have some structural restrictions to ensure generalization to messages unseen during the training [40]. Applying parameter-sharing sequential neural models such as CNN and RNN are natural parametrization methods for both the encoding and the decoding blocks.

RNN models such as Gated Recurrent Unit (GRU) and Long-Short Term Memory (LSTM) are commonly used for sequential modeling problems [41]. RNN is widely used in deep learning based communications systems [23] [31] [24] [35], as RNN has a natural connection to sequential encoding and decoding algorithms such as convolutional code and BCJR algorithm [23].

However RNN models are: (1) of higher complexity than CNN models, (2) harder to train due to gradient explosion, and (3) harder to run in parallel [32]. In this paper, we use one dimensional CNN (1D-CNN) as the alternative encoding and decoding model. Although the longest dependency length is fixed, 1D-CNN has lower complexity, better trainability [42], and easier implementation in parallel via AI-chips [43]. The learning curve comparison between CNN and RNN is shown in Figure 4 left. Training CNN-based model converges faster and more stable than RNN-based GRU model. The TurboAE complexity is shown in appendix.

**Power Constraint Block**: The operation of power constraint blocks (i.e., $h(\cdot)$ in $x = h(b)$) depends on the requirement of power constraint.

Soft power constraint normalize the power of code, as $E(x) = 0$ and $E(x^2) = 1$. TurboAE-continuous with soft power constraint allows the code $x$ to be continuous. Addressing the statistical estimation issue given a limited batch size, we use normalization method [44] as:$x_i = \frac{b_i - \mu(b)}{\sigma(b)}$, where $\mu(b) = \frac{1}{K} \sum_{i=1}^{K} b_i$ and $\sigma(b) = \sqrt{\frac{1}{K} \sum_{i=1}^{K} (b_i - \mu(b))^2}$ are scalar mean and standard deviation estimation of the whole block. During the training phase, $\mu(b)$ and $\sigma(b)$ are estimated from the whole batch. On the other hand, during the testing phase, $\mu(b)$ and $\sigma(b)$ are pre-computed with multiple batches. The normalization layer can be also considered as BatchNorm without affine projection, which is critical to stabilize the training of the encoder [45].

## 3.2 Design of TurboAE-binary – Binarization via Straight-Through Estimator

Some wireless communication system requires a hard power constraint, where the encoder output is binary as $x \in \{-1, +1\}$ [46] - so that every symbol has exactly the same power and the information is conveyed in the sign. Hard power constraint is not differentiable, since restricting $x \in \{-1, +1\}$ via $x = \text{sign}(b)$ has zero gradient almost everywhere. We combine normalization and Straight-Through Estimator (STE) [47] [48] to bypass this differentiability issue. STE passes the gradient of $x = \text{sign}(b)$ as $\frac{\partial x}{\partial b} = \mathbb{1}(|b| \leq 1)$ and enables training of an encoder by passing estimated gradients to the encoder, while enforcing hard power constraint.

Simply training with STE cannot learn a good encoder as shown in Figure 4 right. To mitigate the trainability issue, we apply pre-training, which pre-trains TurboAE-continuous firstly, and then add the hard power constraint on top of soft power constraint as $x = \text{sign}(\frac{b - \mu(b)}{\sigma(b)})$, whereas the gradient is estimated via STE. Figure 4 right shows that with pre-training, TurboAE-binary reaches Turbo performance within 100 epochs of fine-tuning.

TurboAE-binary is slightly worse than TurboAE-continuous as shown in Figure 1, especially at high SNR, since: $(a)$ TurboAE-continuous can be considered as a joint coding and high order modulation scheme, which has a larger capacity than binary coding at high SNR [46], and $(b)$ STE is an estimated gradient, which makes training encoder more noisy and less stable.

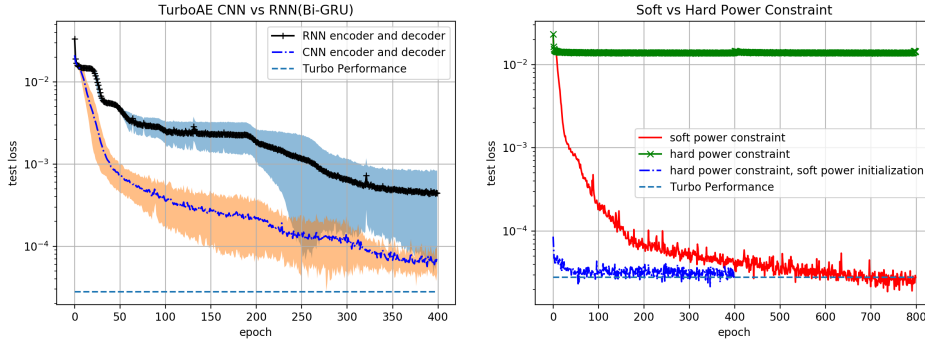

Figure 4: Learning Curves on CNN vs GRU: CNN shows faster training convergence (left); Training with STE requires soft-constraint pre-training (right)

## 3.3 Neural Trainability Design

The training algorithms for training TurboAE are shown in Algorithm 1. Compared to the conventional deep learning model training, training TurboAE has the following differences:

- **Very Large Batch Size** Large batch size is critical to average the channel noise effects. Empirically, TurboAE reaches Turbo performance only when the batch size is grater than 500.

- **Train Encoder and Decoder Separately** We train encoder and decoder separately as shown in Algorithm 1, to avoid getting stuck in local optimum [27] [35].

**Algorithm 1** Training Algorithm for TurboAE

---

**Require:** Batch Size $B$, Train Encoder Steps $T_{enc}$, Train Decoder Steps $T_{dec}$, Number of Epoch $M$
    Encoder Training SNR $\sigma_{enc}$, Decoder Training SNR $\sigma_{dec}$
    **for** $i \leq M$ **do**
        **for** $j \leq T_{enc}$ **do**
            Generate random training example $u$, and random noise $z \sim N(0, \sigma_{enc})$.
            Train encoder $f_\theta$ with decoder fixed, with $u$ and $z$.
        **end for**
        **for** $j \leq T_{dec}$ **do**
            Generate random training example $u$, and random noise $z \sim N(0, \sigma_{dec})$.
            Train decoder $g_\phi$ with encoder fixed, with $u$ and $z$.
        **end for**
    **end for**

---

- **Different Training Noise Level for Encoder and Decoder** Empirically, while it is best to train a decoder at a low training SNR as discussed in [23], it is best to train an encoder at a training SNR that matches the testing SNR, e.g training encoder at 2dB results in good encoder when testing at 2dB [35]. In this work, we use random selection of -1.5 to 2 dB for training the decoder, and test and train the encoder at the same SNR.

We do a detailed analysis of training algorithms in the supplementary materials. The hyper-parameters are shown in Table 1.

| Loss | Binary Cross-Entropy (BCE) |
|---|---|
| Encoder | 2 layers 1D-CNN, kernel size 5, 100 filters for each $f_{i,\theta}(.)$ block |
| Decoder | 5 layers 1D-CNN, kernel size 5, 100 filters for each $g_{\phi_{i,j}(.)}$ block |
| Decoder Iterations | 6 |
| Info Feature Size F | 5 |
| Batch Size | 500 when start, double when saturates for 20 epochs, till reaches 2000 |
| Optimizer | Adam with initial learning rate 0.0001 |
| Training Schedule for Each Epoch | Train encoder $T_{enc} = 100$ times, then train decoder $T_{dec} = 500$ times |
| Block Length K | 100 |
| Number of Epochs M | 800 |

Table 1: Hyper-parameters of TurboAE

## 4 Experiment Results

### 4.1 Block length coding gain of TurboAE

As block length increases, better reliability can be achieved via channel coding, which is referred to as *blocklength gain* [11]. We compare TurboAE (only TurboAE-continuous is shown in this section) with the Turbo code and CNN-AE, tested at BER at 2dB on different block lengths, shown in Figure 5 left. Both CNN-AE and TurboAE are trained with block length 100, and tested on various block lengths. As the block length increases, CNN-AE shows saturating blocklength gain, while TurboAE and Turbo code reduce the error rate as the block length increases. Naively applying general purpose neural network such as CNN to channel coding problem cannot gain performance on long block lengths.

Note that TurboAE is still worse than Turbo when the block length is large, since long block length requires large memory usage and more complicated structure to train. Improving TurboAE on very long block length remains open as an interesting future direction.

The BER performance boosted by neural architecture design is shown in Figure 5 right. We compare the fine-tuned performance among CNN-AE, TurboAE, and TurboAE without interleaving as $x^\pi = \pi(x)$. TurboAE with interleaving significantly outperforms TurboAE without interleaving and CNN-AE.

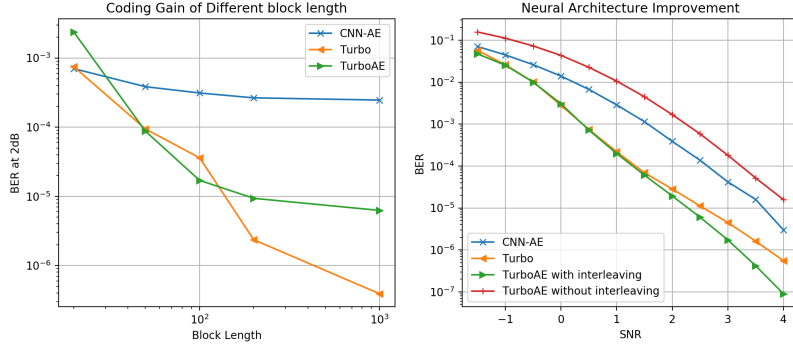

Figure 5: Interleaving improves blocklength gain (left); Neural Architecture improves BER performance (right).

## 4.2 Performance on non-AWGN channels

Typically there are no close-form solutions under non-AWGN and non-iid channels. We compare two benchmarks: (1) canonical Turbo code, and (2) DeepTurbo Decoder [24], a neural decoder fine-tuned at the given channel. We test the performance on both iid channels and non-iid channels in settings as follows:

($a$) iid Additive T-distribution Noise (ATN) Channel, with $y_i = x_i + z_i$, where iid $z_i \sim T(\nu, \sigma^2)$ is heavy-tail (tail weight controlled based on the parameter $\nu = 3.0$) T-distribution noise with variance $\sigma^2$. The performance is shown in Figure 6 left.

($b$) non-iid Markovian-AWGN channel, is a special AWGN channel with two states, {good, bad}. At bad state the noise is worse by 1dB than the SNR, and at good state, the noise is better by 1dB than the SNR. The state transition probability between good and bad states are symmetric as $p_{bg} = p_{gb} = 0.8$. The performance is shown in Figure 6 right.

For both ATN and Markovian-AWGN channels, DeepTurbo outperforms canonical Turbo code. TurboAE-continuous with learnable encoder outperforms DeepTurbo in both cases. TurboAE-binary outperforms DeepTurbo on ATN channel, while on Markovian-AWGN channel, TurboAE-binary does not perform better than DeepTurbo at high SNR regimes (but still outperforms canonical Turbo). With the flexibility of designing an encoder, TurboAE designs better code than handcrafted Turbo code, for channels without a closed-form mathematical solution.

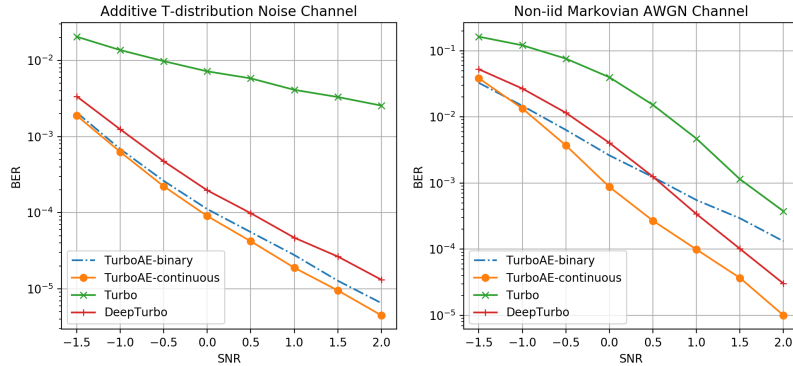

Figure 6: TurboAE on iid ATN channel (left) and on-iid Markovian-AWGN channel (right)

# 5 Conclusion and discussion

In summary, in this paper, we propose TurboAE, an end-to-end learnt channel coding scheme with novel neural structure and training algorithms. TurboAE learns capacity-approaching code on various channels under moderate block length by building upon 'turbo principle' and thus, exhibits discovery of codes for channels where a closed-form representation may not exist. TurboAE, hence, brings an interesting research direction to design channel coding algorithms via joint encoder and decoder design.

A few pending issues hamper further improving TurboAE. **Large block length** requires extensive training memory. With enough computing resources, we believe that TurboAE's performance at larger block lengths can potentially improve. **High SNR** training remains hard, as in high SNR the error events become extremely rare. **Optimizing BLER** requires novel and stable objective for training. Such pending issues are interesting future directions.

### Acknowledgments

This work was supported in part by NSF awards 1908003, 651236 and 1703403.

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
