[Supplementary Material]

# Supplementary Materials for Turbo Autoencoder: Deep learning based channel code for point-to-point communication channels

## 1  TurboAE Design Analysis

### 1.1  Neural Architecture Design

#### 1.1.1  Supporting code rates beyond 1/3

In main text, only neural code for code rate $R = 1/3$ is shown. The TurboAE encoder and decoder for code rate 1/2 is shown in Figure 1. Still designed under 'Turbo principle', TurboAE with code rate 1/2 shows impressive performance under low to moderate SNR, within block length 100. To generate code rates beyond 1/2, we can utilize puncturing.

Figure 1: The encoder structure (up left), decoder structure (down left), and BER performance (right) of code rate 1/2

#### 1.1.2  CNN with Residual Connection

The same shape property of 1D-CNN is preserved by setting odd kernel size $k$ equals twice the zero-padding length minus one, as shown in Figure 2 left. The encoder simply use 1D-CNN as encoder blocks, while the decoder uses residual connection to bypass gradient on iterative decoding procedure to improve trainability [1], and also inspired by extrinsic information from Turbo code [2], shown in Figure 2 right. Adding residual connection improves training speed and improve final BER performance [3].

#### 1.1.3  Network Size

In figure 3 left, we show the test loss trajectory of TurboAE with different network size. We keep both encoder and decoder with same number of filters. Larger network lead to faster training and better performance, with the cost of larger computation and memory usage. We take encoder and

Figure 2: 1D CNN visualization on 1 layer (left); CNN with residual connection (right).

decoder with 100 filters, which trains fast given limited computational resource (e.g., training 400 epochs takes 1 day on one Nvidia 1080Ti.)

Figure 3: Larger Network has better performance (left); Random interleaving array shows same performance (right).

### 1.1.4 Random Interleaving Array During Testing Phase

Given a fixed pseudo-random interleaving array, one concern is that TurboAE could overfit to specific interleaving array, and when both encoder and decoder change the interleaving array, TurboAE will have a degraded performance. However, empirically, we observe that TurboAE doesn't overfit to the training fixed pseudo-random interleaving array, as shown in Figure 3 right. The TurboAE is trained on one specific interleaving array, and tested on 3 random generated interleaving arrays. For TurboAE, whenever the interleaving array is pseudo-random, the neural encoder and decoder still learn without overfitting.

However, when the interleaving array is not random, e.g not applying interleaving as $y = \pi(x)$, termed as 'no interleaving', the performance degrades significantly.

### 1.2 Training Algorithms

### 1.2.1 Joint Training vs Separate Training

Empirically training encoder and decoder simultaneously is easier to get stuck in local optimum as shown in Figure 4 left. Training encoder and decoder separately is less likely to get stuck in local optimum [4] [5]. Training decoder more times than encoder, on the other hand, makes decoder better approximates optimal decoding algorithm of the encoder, which offers more accurate estimated gradient and stabilizes the training process [5]. We training encoder and decoder separately, with each epoch trains encoder 100 times and decoder 500 times.

### 1.2.2 Large Batch Size Improves Training Significantly

Large batch size helps training deep generative models such as Generative Adversarial Networks (GAN) [6] and Variantional Autoencoder (VAE) [7], and is also critical to training TurboAE. Figure 4 right shows that large batch size leads to significantly lower test BER.

Figure 4: Training encoder and decoder jointly gets stuck as local optimum (left). Large batch size improves training (right).

The analysis is on AWGN channel by using the 1st order Taylor expansion on decoder $g_\phi(.)$ as:
$\hat{u} = g_\phi(x + z) = g_\phi(x) + zg'_\phi(x) + O(z^2)$.

Taking gradient of both sides becomes: $\frac{\partial \hat{u}}{\partial x} \approx g'_\phi(x) + zg''_\phi(x)$ and $\frac{\partial \hat{u}}{\partial \phi} \approx \frac{\partial g_\phi(x)}{\partial \phi} + z\frac{\partial g'_\phi(x)}{\partial \phi}$

AWGN channel has $\frac{\partial y}{\partial x} = 1$ with iid noise. Consider the normalization layer $x = h(b)$, the gradient pass through normalization layer with batch size $B$ is [8]:

$$\frac{\partial x_i}{\partial b_j} = \frac{1}{\sigma(b)}(\mathbb{1}(i = j) - \frac{1}{B}(1 + b_i b_j)) \tag{1}$$

Known $\hat{u} = sigmoid(q)$, as $q = g_\phi(h(f_\theta(u)) + z)$, the gradient of BCE loss with respect to logit $q$ is $\frac{\partial BCE(u,\hat{u})}{\partial q} = \hat{u} - u$, the gradient of encoder is:

$$\frac{\partial BCE(u,\hat{u})}{\partial \theta} = \frac{\partial BCE(u,\hat{u})}{\partial q}\frac{\partial q}{\partial y}\frac{\partial y}{\partial x}\frac{\partial x}{\partial b}\frac{\partial b}{\partial \theta} = (\hat{u} - u)(g'_\phi(x) + zg''_\phi(x))\frac{\partial x}{\partial b}\frac{\partial f_\theta(u)}{\partial \theta} \tag{2}$$

The gradient of decoder is:

$$\frac{\partial L}{\partial \phi} = \frac{\partial BCE(u,\hat{u})}{\partial q}\frac{\partial q}{\partial \phi} = (\hat{u} - u)(\frac{\partial g_\phi(x)}{\partial \phi} + z\frac{\partial g'_\phi(x)}{\partial \phi}) \tag{3}$$

The benefits of large batch size are as follows:

- **Less noisy gradient for encoder.** The gradient passes through normalization layer is as shown in Equation (1). With large batch size $B$, the gradient passes through normalization reduces the noise introduced by $\frac{1}{B}(1 + b_i b_j)$, making the gradient passed to encoder less noisy.

- **Larger batch size reduces gradient noise.** Large batch size makes gradient estimation for both encoder and decoder more accurate, as the error term $zg''_\phi(x)$ in Equation (2) and $z\frac{\partial g'_\phi(x)}{\partial \phi}$ in Equation (3) can be reduced with large batch size with expectation $E[z] = 0$. Better gradients for both encoder and decoder improve training.

- **More accurate statistics for normalization.** With larger batch size, the mean and the standard deviation for normalization used in power normalization are more accurate, which introduces less noise.

### 1.2.3 Training SNR

Training noise level (SNR) is an critical parameter for training TurboAE. The training SNR analysis can be derived by the gradient analysis of section 1.2.2. The training noise has different effect on

encoder and decoder. The training noise affects decoder with noise term $z\frac{\partial g'_\phi(x)}{\partial \phi}$ in Equation (3). Given an fixed encoder, training decoder with different SNR results in different levels of regularization. For encoder there are two source of noise regularizations: (a) $zg''_\phi(x)$ in Equation (2,) and (b) noise introduced by normalization layer in Equation (1). Training encoder with different SNR also results in different levels of regularization, which differs from training decoders.

As the effect of decoder training noise has been studied in [9], in this section, we study the training SNR of encoder, with fixing decoder training SNR to be 0dB as shown in Figure 5 left. We see that the most reliable code can be learned when training SNR matches testing SNR. Throughout the paper, we make encoder training SNR equals the testing SNR, e.g we testing TurboAE performance at 2dB, we train TurboAE with encoder SNR at 2dB and decoder at 0dB. The BER curve shown in main context is the lower envelope of all curves.

Figure 5: Encoder Training SNR has different coding gain effects (left); Training decoder more lead to faster convergence (right).

When encoder training SNR is larger than 1dB (e.g., 1dB, 2dB and 3dB), the BER curves remain nearly the same. Thus encoder training noise level for high SNR region shows diminishing effects on high SNR, which creates an error floor for TurboAE. In main context we state that neural code are suboptimal in high SNR region, since the error is hard to encounter (with probability less than $10^{-4}$), which makes it hard to gather negative examples to train encoder. Improving high SNR region coding gain with data imbalance is an interesting future research direction.

### 1.2.4 Train decoder more than encoder

We argue that when the decoder is well-paired to the fixed encoder, the gradient passed to encoder is more accurate. Training decoder more times will improve performance, as shown in Figure 5 right. Training decoder more times lead to faster convergence.

### 1.2.5 Learning Rate and Batch Size Scheduling

Increasing batch size improve generalization rather than reducing learning rate [10]. To reduce computational expense, we start with batch size $B = 500$, and double the batch size when the test loss saturates for 20 epochs till $B = 2000$ which is our GPU memory limit. Figure **??** shows that there exists long 'fake saturating' points where the test loss saturates for over 20 epochs and then continue to drop. When $B = 2000$, when saturates for longer than 20 epochs, the learning rate $lr$ is reduced by 10 times till learning rate reaches $lr = 0.000001$.

### 1.2.6 Block Error Rate Performance comparison

The loss function used is Binary Cross-Entropy (BCE), which minimizes average cross entropy for all bits along the block, aiming at minimizing BER. Optimizing BER doesn't necessarily result in optimizing block error rate (BLER), as shown in Figure 6. TurboAE-binary shows better performance comparing to Turbo code in BER sense under all SNR points, the BLER performance is worse than Turbo code.

Figure 6: TurboAE BER (left) and BLER (right) performance

## 2 Complexity Comparison

Neural networks are known to have high implementation complexity than canonical algorithms. CNN structure is more favorable than RNN since it is of less complexity and easier to run in parallel.

We compare the inference complexity between TurboAE with CNN and GRU implementations (with similar performance), as well as canonical Turbo decoder in this section. The neural network computation is measured via float-point operations (FLOP) in one block. Turbo's encoder and decoder complexity is computed in elementary math operations (EMO), which are are in Table 1 :

| Metric | CNN encoder | CNN decoder | GRU encoder | GRU decoder | Turbo encoder | Turbo decoder |
|---|---|---|---|---|---|---|
| FLOP/EMO | 1.8M | 294.15M | 334.4M | 6.7G | 104k | 408k |
| Parameters | 152.4k | 2.45M | 1.14M | 2.714M | N/A | N/A |

Table 1: FLOP and number of parameter comparison on block length 100 and 6 iterations

CNN encoder and decoder are considered as small, comparing to typical deep learning models which take about 1G FLOP per instance. GRU has much larger FLOP comparing to CNN. Empirically using GRU takes 10x GPU memory and is 10x slower to train. However comparing to canonical Turbo encoder and decoder, FLOP of TurboAE with CNN is still much larger than canonical decoders.

We are expecting continuing research would lead to smaller FLOP, as well as the advance of AI-chips will increase the performance when applying CNN to TurboAE.

Due to TurboAE complexity and flexibility, and superior performance on moderate block length on low-to-moderate SNR, the best application area for TurboAE is on dynamical environment (e.g operating on moderate block length and channel with uncertainty) such as low latency code and control plane. On high throughput data plane where canonical codes such as LDPC and Turbo, or neural decoder can be the best method with low complexity and high reliability. In the future, combining both adaptive neural code and human-designed capacity-approaching codes will give more seamless and high reliable communication experience.

## 3 TurboAE Performance

### 3.1 Benchmarks

We use MATLAB-based Vienna 5G simulator and Python-based Commpy [12] as our benchmarks.

#### 3.1.1 Vienna 5G simulator

The detailed implementation details of Vienna 5G are:

- LDPC code with PWL-Min-Sum decoding algorithm, with 32 decoding iterations.

126      • Polar code with CRC-List-SC decoding algorithm, with list size 32.

127      • Turbo code with Linear-Log-MAP decoding algorithm, with 6 decoding iterations.

128      • TBCC code with MAX-Log-MAP decoding algorithm.

129 TurboAE and Turbo code uses the same number of iterations. Turbo codes simulation results are
130 different between Commpy and Vienna 5G simulator, since Commpy implements vanilla Turbo
131 code, and Vienna 5G simulator implements more advanced coding Turbo schemes. We use Commpy
132 Turbo code as our benchmark. Note that Commpy shows better performance than Vienna 5G, but
133 shows less coding gain on high SNR. We use Commpy as the benchmark, which is the same as [9].

134 For Vienna 5G simulator, we find that the code rate for each channel coding is not enforced, e.g
135 when setting code rate $R = 1/3$ with block length $K = 100$, the encoder not necessarily outputs
136 codeword with block length $N = 300$, but rather outputs longer block length $N = 384$. To make
137 a fair comparison, we tune the code rate to enforce the output of encoder to have block length
138 $N = 300$, which results in a different setup code rate:

139      • Polar code: for code rate $R = 1/2$, the setup code rate is $R = 0.62$; for code rate $R = 1/3$,
140        the setup code rate is $R = 0.415$.

141      • TBCC Code: for code rate $R = 1/2$, the setup code rate $R = 0.64$, for code rate $R = 1/3$,
142        the setup code rate $R = 0.4275$.

143      • Turbo Code: for code rate $R = 1/2$, the setup code rate $R = 0.62$, for code rate $R = 1/3$,
144        the setup code rate $R = 0.4175$.

145      • LDPC code: for code rate $R = 1/2$, the setup code rate $R = 0.705$, for code rate $R = 1/3$,
146        the setup code rate $R = 0.522$.

147 Interested reader can contact Vienna 5G simulator's authors to get access to the code.

### 3.1.2   Commpy on Turbo Code

149 RSC code with generating function $(1, \frac{f_1(x)}{f_2(x)})$ is the component code for Turbo code. The generating
150 function of Turbo's RSC affects the performance. Two commonly used configurations of RSC are
151 implemented in Commpy:

152      • code rate $R = 1/3$, with $f_1(x) = 1 + x^2$ and $f_2(x) = 1 + x + x^2$, which is denoted as
153        Turbo-757.

154      • code rate $R = 1/3$, with $f_1(x) = 1 + x^2 + x^3$ and $f_2(x) = 1 + x + x^3$, which is standard
155        Turbo code used in LTE system, denoted as Turbo-LTE.

156 In main context, the benchmarks are using with Turbo-757, while the performance comparison
157 between Turbo-757 and Turbo-LTE are shown in Figure 7. The performance trend are the same,
158 while Turbo-LTE shows slightly better performance. The same claim in main text on Turbo-757,
159 works for Turbo-LTE.

### 3.2   Continuous Channels

161 Continuous channel refers to the channel where the received signal $y$ can is continuous, where both
162 TurboAE-continuous and Turbo-binary can be supported. We discussed short block performance on
163 AWGN and non-AWGN channels, and in this section we discuss the longer block length, and other
164 channels.

### 3.2.1   Scale to Long Code Block Length is hard

166 Figure 8 left shows that after fine-tuning at block length 1000, fine-tuned TurboAE shows improved
167 performance comparing to TurboAE trained on block length 100 and tested on block length 1000.
168 However, TurboAE-continuous shows worse performance comparing to canonical Turbo code. As
169 shown in main context, TurboAE's coding gain on long block length is smaller than Turbo code due
170 to trainability and computation issues. Improving performance on long block length is an interesting
171 future research direction.

Figure 7: Commpy TurboAE with different trellis performance BER (left), and BLER (right)

### 3.2.2 TurboAE on iid Rayleigh Fading Channel

Non-coherent Rayleigh Fading Channel is defined as $y_i = h_i x_i + z_i$, where iid $z_i \sim N(0, \sigma^2)$, and $h_i$ is normalized iid Rayleigh distribution fading noise as $h_i \sim \frac{\sqrt{U^2 + V^2}}{\sqrt{\pi/2}}$, while $U$ and $V$ are IID unit Gaussian variables. Non-coherent setting means the decoder doesn't know $h_i$: the benchmarks (canonical decoders include Turbo, TBCC, and LDPC) are not aware of the fading component by still taking log-likelihood as decoder input, while TurboAE is not further trained to learn $h_i$. The performance of Non-coherent Rayleigh Fading Channel are shown in Figure 8 right. On Non-coherent Rayleigh Fading Channel, TurboAE-binary and TurboAE-continuous outperforms LDPC, TBCC and Turbo code in a wide SNR region.

Figure 8: TurboAE performance on block length 1000 (left) and TurboAE on Rayleigh Fading Channel (right)

### 3.2.3 TurboAE-continuous combines Modulation and Coding

In the main context, we show that on ATN channel, TurboAE-continuous outperforms TurboAE-binary. TurboAE-continuous outperforms TurboAE-binary significantly at high SNR since TurboAE-continuous jointly learns modulating and coding in continuous value domain, which has better advantage at high SNR schemes.

To investigate the fundamental coding gain of TurboAE-continuous in high SNR schemes, we investigate the channel capacity of non-AWGN channel (take ATN as example) and AWGN channel, as shown in Figure 9 right. Binary-AWGN and Continuous-AWGN refers to the channel capacity where code $x$ is binary and continuous at AWGN channel, respectively. Binary-ATN and Continuous-ATN refers to the channel capacity where code $x$ is binary and continuous at ATN chan-

nel. We use estimated Mutual Information (MI) via KSG estimator [13] as the surrogate measure
for channel capacity, as there is no close-form channel capacity for ATN channel.

Under the SNR range where most channel coding operates around (0dB), the MI between Binary-
AWGN and Continuous-AWGN is very close, thus applying continuous coding doesn't improve cod-
ing gain significantly on AWGN channel. However, the MI between binary-ATN and Continuous-
ATN is significant, thus applying continuous code can further increase the channel capacity compar-
ing to using binary code on non-AWGN channel. Moreover, at high SNR, the capacity of continuous
code is much larger than binary code, which shows that Turbo-AE, as a method to learn continuous
code, has theoretical advantage on high SNR schemes.

Figure 9: KSG estimated Mutual Information for AWGN and ATN channel

## 3.3 Binary Channels

Binary channels restrict the decoder input to be binary, which only supports binary operations. Only
TurboAE-binary is supported. We use the following canonical binary channels:

- iid Binary Symmetric Channel (BSC), $x \in \{-1, +1\}$ and $y \in \{-1, +1\}$, flip rate $P(y \neq x) = p_{bsc}$, and $P(y = x) = 1 - p_{bsc}$.

- iid Binary Erasure Channel(BEC), $x \in \{-1, +1\}$ and $y \in \{-1, 0, +1\}$, while $y = 0$ represents erasure. Erasure rate $P(y = 0) = p_{bec}$, and $P(y = x) = 1 - p_{bec}$.

Figure 10: BEC and BSC performance

On BSC channel, TurboAE-binary and Turbo works nearly the same, which implies that AWGN-
trained TurboAE can generalize to BSC channel.

On BEC channel, TurboAE-binary trained on AWGN works worse than Turbo, since on AWGN channel there doesn't exist erasure. However TurboAE fine-tuning on BEC channel still has gap comparing to Turbo. The result shows that the trainability of TurboAE still needs improvement.

## 3.4 Interleaved Encoding Visualization

We test the random coding effect of interleaved encoder with 2 same message, $u_1$ and $u_2$, and perturb at position index 20, which makes the only difference between $u_1$ and $u_2$ is $s_1[20] = 1.0$ and $s_1[20] = 0.0$. We plot the absolute maximized code difference $|f_\theta(s_1) - f_\theta(s_2)|$ for all three encoding blocks. With interleaved encoder, one single message bit change (at code bit 20) can cause random non-adjacent bits (at code bit 75) to change, which adds encoding randomness, shown in Figure 11.

Figure 11: Randomness added via interleaving