[Reviews · NeurIPS 2019]

Reviewer 1



This paper proposes the first channel coding schemes with both encoder and decoder designed through neural networks. The code seems to outperform the state-of-the-art coding scheme in certain scenarios, and also has good generalization property to other channels. It also suggests some training guidelines, which is crucial for obtaining a good channel code. My only concern is the complexity of the code designed in such a way. In case it ends up having exponential complexity in the block length, perhaps it is unfair to compare it to the state-of-the-art low complexity coding scheme. The right target, in that case, should be how close it gets to the performance of the maximum likelihood decoding. Nevertheless, the idea in this paper is original and the writing is clear and easy to understand. After rebuttal: In the authors' feedback, they claim that complexity analysis will be added in the final version. I look forward to reading this part. I also read the authors' response to other reviewers and felt it is reasonable. Therefore, I am in favour of acceptance.

Reviewer 2



In recent years, several papers have employed deep learning methods to decode various classes of codes (turbo codes, linear codes, polar codes). This work focuses on turbo codes, and has the more ambitious goal of providing joint training of the decoder and the encoder (which means that the resulting code will not be a turbo code in the traditional sense). The authors borrow some ideas from the turbo coding literature (e.g., interleaving) and use CNNs to design decoder and encoder (as opposed to RNNs used in several other papers). The proposed TurboAE algorithm achieves performance which is comparable to state-of-the-art codes (see Figure 1). This is quite impressive, even though the code length is quite short (i.e. 100 bits). In fairness, this is common issue as deep learning techniques tend not to scale well with the block length. The authors also consider non-AWGN channels and compare with the original turbo decoder and with DeepTurbo proposed in [26]. Overall, the paper is interesting, clear and well-written. I have the feeling that the authors emphasize a bit too much their results (which are quite good anyway), and some detailed comments are in the 'Improvements' section.

Reviewer 3



The idea of turbo AE is novel as far as I know, and it is interesting. It allows to learn a code of long length. The alternative training process presented in this paper enable us to train both an encoder and a decoder simultaneously. Unfortunately, from the experimental results, advantages (scalability, comparison to the conventional turbo codes) of the proposed scheme appears not so clear. I increased the overall score to 7 from 4.

[Author Response · NeurIPS 2019]

We thank the reviewers for their insightful comments. Please find the detailed responses below.

**Response to Reviewer 1.**

**Complexity analysis:** TurboAE has a linear complexity in the block length, in runtime and computation. We will
provide the exact comparison in the revised version. The run time comparison is non-trivial because TurboAE can be
run on GPUs and AI-chips which can highly parallelize the computations. We will provide timing experiments for
TurboAE and traditional turbo codes run on both GPU and CPU in the revision.

**Response to Reviewer 2.**

**1. Decoder design for non-AWGN:** We agree with the reviewer that the statement is confusing and misleading. We
just meant to point out that there are heuristics used in practice that leaves room for improvement. We will revise the
sentences to make it clear.

**2. Comparison to polar and LDPC codes:** We will add more figures for comparisons in the revised version. To
mention the results briefly,
2-1. For continuous non-AWGN (ATN) channels, the reliability of polar and LDPC codes are very similar to turbo
code; TurboAE outperforms polar and LDPC codes by a large margin.
2-2. For binary input channels, TurboAE is comparable to turbo, polar, and LDPC for BSC and is slightly worse than
these traditional codes under BEC (Figure 10, Supplementary 3.3). While TurboAE does not outperform traditional
codes, achieving a comparable reliability under *discrete* channels requires a major breakthrough. This is because
learning an autoencoder including a non-differentiable layer (binarization of code) makes training challenging. We
resolve this issue via Straight-Through Estimator.

**3. Generalization to longer block lengths:** As noted, it is a common issue that deep learning techniques tend not
to scale well with the block length. Up until now, deep learning based codes were shown to meet state-of-the-art for
AWGN channel for very short block length. By applying the interleaving idea, we can achieve state-of-the-art reliability
for block length 100. "Can we go beyond this?" is certainly a central question. A short answer is "it is not easy." We
show the result of direct generalization to longer block lengths (without re-training) in Figure 5 left; as block length
increases, BER decreases slowly. However, we want to emphasize that we are the first to achieve such block length
gain (error rate decreasing with increasing block lengths) with a neural networks based encoder. And this is achieved
without re-training at the larger block-length.
How to increase this block-length gain is an interesting future research direction. Re-training at the larger block-length
is slow, but the reliability is improved (Appendix Section 3.2. Figure 8). However, it is still not comparable to traditional
turbo codes at that larger block length.

**4. Comparison to normal approximation and other codes (Liva et al.):** TurboAE meets normal approximation at
low SNR (< 2dB) but is worse than normal approximation (and other codes) at high SNRs (> 2dB); we will include the
comparison to Liva et al.'s baselines in the revised version. This pattern matches with the pattern in Figure 1, which
shows that TurboAE's performance on high SNR is worse than LDPC and Polar codes. We conjecture that it is because,
for high SNR regimes, almost all examples shown in the training are easy to label, as they have small noise. We see
(relatively) less examples at the decision boundaries, making it hard to train an accurate decoder. Overcoming this
challenge is left as a future work. For example, an adversarial training can be used.

**5. Comparison to RNN-decoders:** We compare TurboAE with DeepTurbo (which is RNN-based decoder) as it
achieves state-of-the-art reliability in decoding turbo codes; DeepTurbo outperforms both traditional decoder and
CNN-based decoders [26]. In Figure 6 (Section 4.2), we show that TurboAE outperforms DeepTurbo for ATN and
AWGN channels with memory. For AWGN channel, TurboAE is comparable to DeepTurbo. We will include this
comparison in the modified version.

**Response to Reviewer 4.**

**Advantage of TurboAE over existing codes:** It is true that under AWGN channels, TurboAE achieves reliability
comparable to the state-of-the-art. We would like to reiterate (1) why this is interesting, and (2) advantages of TurboAE
over existing codes:

1. For AWGN channels, existing codes are already very close to optimal, but it took 50 years of mathematicians' efforts
to derive these codes. So the central question is not necessarily whether we can beat these existing codes on AWGN, but
rather whether we can discover completely new codes, automatically from data. The major contribution of TurboAE is
to demonstrate an alternative neural network based approach to discover new codes in a data-driven manner.

2. For several channels beyond AWGN, existing codes can be far from optimal. We demonstrate that for those channels
(e.g., ATN, AWGN with memory), TurboAE outperforms existing codes by a large margin (e.g., 5–10x in reliability).
Hence, our work defines a new family of possibilities in code and decoder design for open problems in coding theory.

[Meta-Review · NeurIPS 2019]

This paper considers a combination of encoder and decoder architecture, which is a serially-concatenated code with interleavers, as in the turbo codes, combined with turbo-like iterative decoding, and proposes implementing encoders and decoders of the constituent codes with 1D-CNN, which allow us to train the encoders and the decoders in an end-to-end and data-driven fashion. Two reviewers raised concern about the scalability issue of the proposal, and the authors admit in their rebuttal that it is a central question. Although the review scores exhibited a large split in the initial round of review, mainly due to the scalability issue as well as comparison in performance with other existing coding schemes, after the authors' rebuttal all the reviewers rated this paper above the acceptance threshold. I would therefore like to recommend acceptance of this paper.